# The Relevance of Dynamical Friction for the MW/LMC/SMC Triple System

**Wolfgang Oehm** [1,2,*] and **Pavel Kroupa** [2,3,*]

1 scdsoft AG, Albert-Nestler-Str. 21, 76131 Karlsruhe, Germany
2 Helmholtz-Institut für Strahlen und Kernphysik (HISKP), Nussallee 14-16, 53115 Bonn, Germany
3 Astronomical Institute, Faculty of Mathematics and Physics, Charles University in Prague, V Holešovičkách 2, 18000 Prague, Czech Republic
* Correspondence: physik@wolfgang-oehm.org (W.O.); pkroupa@uni-bonn.de (P.K.)

**Abstract:** Simulations of structure formation in the standard cold dark matter cosmological model quantify the dark matter halos of galaxies. Taking into account dynamical friction between dark matter halos, we investigate the past orbital dynamical evolution of the Magellanic Clouds in the presence of the Galaxy. Our calculations are based on a three-body model of rigid Navarro–Frenk–White profiles for dark matter halos but were verified in a previous publication by comparison to high-resolution $N$-body simulations of live self-consistent systems. Under the requirement that the LMC and SMC had an encounter within 20 kpc between 1 and 4 Gyr ago in order to allow the development of the Magellanic Stream, using the latest astrometric data, the dynamical evolution of the MW/LMC/SMC system is calculated backwards in time. With the employment of the genetic algorithm and a Markov-Chain Monte-Carlo method, the present state of this system is unlikely, with a probability of $< 10^{-9}$ ($6\sigma$ complement), because the solutions found do not fit into the error bars for the observed plane-of-sky velocity components of the Magellanic Clouds. This implies that orbital solutions that assume dark matter halos, according to cosmological structure formation theory, to exist around the Magellanic Clouds and the Milky Way are not possible with a confidence of more than 6 sigma.

**Keywords:** galaxies: halos; galaxies: interactions; galaxies: kinematics and dynamics; galaxies: Magellanic Clouds





## 1. Introduction

The standard $\Lambda$CDM cosmological model [1–3] requires about 81 per cent of the matter content of the Universe to comprise pressureless cold dark matter (CDM) particles, which are not described by the standard model of physics, and about 68 per cent of the total mass–energy density of the Universe to be composed of dark energy, represented by the cosmological constant $\Lambda$ [4,5]. The related $\Lambda$WDM model is based on dark matter (DM) being made of warm dark matter particles that have a smaller mass than CDM particles but lead to very similar DM halos within which the observed galaxies reside (e.g., [6–9]). Detecting DM particles has become a major worldwide effort, which has so far yielded null results (e.g., [10–15] and references therein). Finding evidence for DM particles under the assumption that they have a finite interaction cross-section with standard-model particles may lead to null detection if the interaction cross-section is too small to be measurable. However, arguments based on the observed strong correlations between standard particles and dark matter suggest a significant cross-section [16]. This indicates an impasse in the experimental verification of the existence of DM particles.

An alternative and robust method for establishing the existence of DM particles is to develop techniques that rely solely on their gravitational interaction with ordinary matter. A suitable approach is given by Chandrasekhar dynamical friction (e.g., [5]): when a satellite galaxy with its own DM halo enters the DM halo of a host galaxy, its orbit decays as a result of dynamical friction, and the satellite merges with the host. The decay of the orbit does

not depend on the mass of the DM particle but only on the mass density of DM particles, which is fixed by the cosmological parameters. This is the primary reason for the existence of the merger tree in standard cosmology (e.g., [17,18]) and for interacting galaxies to be thought of as merging galaxies. Effectively, a DM halo, being 10–20 times more extended and 50–100 times as massive as the observable part of a galaxy, works like a spider's web. In contrast, if there were to be no DM halos around galaxies, then galaxy–galaxy encounters would be significantly less dissipative, galaxies would encounter each other multiple times, and mergers would be rare. In the eventuality that DM halos were not to exist, however, non-relativistic gravitational theory would need to be non-Newtonian, and given the correlations that galaxies are observed to obey, this theory would need to be effectively Milgromian [19–22]. Galaxy–galaxy encounters and mutual orbits in Milgromian dynamics and without DM halos have been studied (for example, the interacting Antennae galaxy pair [23], and the Milky Way–Andromeda binary explaining the mutually correlated planes of satellites around both hosts [24–26]). The law of gravitation fundamentally defines the formation and evolution of galaxies. For example, by mergers being much rarer in Milgromian gravitation (i.e., without DM), galaxies evolve largely in isolation [27–29], and the formation of cosmological structures proceeds differently than in the DM-based models [5,30,31]. It is thus of paramount importance to test for the existence of DM halos around galaxies.

The test based on Chandrasekhar dynamical friction for the existence of DM halos was introduced by Angus, Diaferio and Kroupa [32] by addressing the question of whether the present-day Galactocentric distances and motion vectors of observed satellite galaxies of the Milky Way (MW) conform to their putative infall many Gyr ago. The solutions for those satellite galaxies for which proper motions were available imply tension with the existence of DM halos since no infall solutions were found. In contrast, without the DM component, the satellite galaxies would be orbiting about the MW, having been most likely born as tidal dwarf galaxies during the MW–Andromeda encounter about 10 Gyr ago [25,26,33]. A further test for the presence of the DM component applying dynamical friction was achieved by Roshan et al. [34], who found the observed bars of disk galaxies to be too long and rotating too fast in comparison with the theoretically expected bars in the presence of DM halos that absorb the bar's angular momentum. The reported discrepancy amounts to significantly higher than the 5-sigma threshold, such that the observations are incompatible with the existence of DM halos. Another independent application of the dynamical friction test is available on the basis of the observed distribution of matter within the M81 group of galaxies. The extended and connected tidal material implies multiple past close encounters of the group members. But the dynamical evolution of the M81 group of galaxies is difficult to understand theoretically if the galaxies are contained in the DM halos that are expected in standard cosmology ([35] and references therein). The problem is that, in the presence of DM halos, the group merges too rapidly to allow the tidal material to be dispersed as observed.

The interesting aspect of these results is that the three independent analyses of the orbital dynamics of the MW satellite galaxies, galactic bars and the M81 group agree in the conclusion that the data are difficult if not impossible to understand in the presence of DM halos. Because the implications of the non-existence of DM bears major implications for theoretical cosmology and galaxy formation and evolution, further tests are important to ascertain the above conclusions, or indeed to question them.

Given the high-accuracy and high-precision position and velocity data available today through the astrometric Gaia mission, the problem of how the existence of the massive and extended DM halos in which galaxies reside can be tested for is revisited here using the process of Chandrasekhar dynamical friction on the triple-galaxy system comprising the MW and the Large (LMC) and Small (SMC) Magellanic Clouds, including the Magellanic Stream. This system is comparable to the M81 system, as it too has a large gaseous structure stemming from the Magellanic Clouds, the Magellanic Stream, which constrains the past orbits of the components of the system. In the following, we assume that the standard

cosmological model is correct and that the MW, the LMC and the SMC are each contained in DM halos that are consistent with those obtained from the theory of structure formation in the ΛCMD model of cosmology. That is, we associate the baryonic mass of each galaxy with a DM halo profile as predicted by the theory in order to test the theory.

Given that the LMC and SMC are about 50 and 60 kpc distant from the centre of the MW and about 20 kpc distant from each other and that the radii of the DM halos are such that all three galaxies are immersed in the dark matter halo of the MW and that the SMC is immersed in the dark matter halo of the LMC and vice versa, it is apparent that Chandrasekhar dynamical friction is likely to play a very significant role in establishing the orbits of the three galaxies relative to each other. Recent work [36–41] suggests the Magellanic Clouds to be on their first pericentre passage such that the bulk properties of their putative DM halos will not be significantly stripped. Detailed dynamical modelling of the Magellanic Clouds problem has, until now, not come to the conclusion that there is a problem.

For example, Besla et al. [37] performed simulations of the LMC and SMC in the first-infall scenario by assuming that dynamical friction from the MW DM halo can be neglected. This is not correct and significantly helps the LMC/SMC pair to exist longer before merging. Indeed, Lucchini et al. [42] studied the formation of the Magellanic Stream using hydrodynamical simulations in the first-infall scenario for the LMC and SMC. Their results, obtained in fully live DM halos of all components such that dynamical friction is correctly computed self-consistently, show that the LMC/SMC binary orbit shrinks significantly more rapidly (their Figure 1) than calculated by Besla et al. [37]. But Lucchini et al. adopted DM halo masses of the LMC ($1.8 \times 10^{11}$ $M_\odot$), of the SMC ($1.9 \times 10^{10}$ $M_\odot$) and of the MW ($1.0 \times 10^{12}$ $M_\odot$) that are significantly less massive than predicted by the ΛCDM model (see Table 1 below). Their calculations also do not consider whether the infalling LMC/SMC binary would have survived for sufficiently long before infall, because it is implausible for the present-day LMC/SMC binary to have formed during or shortly before infall, and whether the infall velocity vector is consistent with the Hubble flow 3.46 Gyr ago at the start of their simulation. Vasiliev [41] studied a scenario in which the LMC is on its second passage past the MW but ignored the SMC. The calculated orbits of the LMC decay due to dynamical friction consistently to the results presented here. Thus, this past work either neglects important contributions to dynamical friction or members in the problem or uses light-weight DM halos that are not consistent with the ΛCDM cosmological model. Essentially, past work has demonstrated that the MW/LMC/SMC triple system can broadly be understood in the context of orbital dynamics by effectively suppressing Chandrasekhar dynamical friction, but it does not demonstrate that the calculated orbits are consistent with the standard DM-based cosmological theory.

In order to assess whether the standard DM-based models can account for the existence of the observed MW/LMC/SMC triple system, the present work studies their orbital history by strictly constraining the DM halo of each of the three involved galaxies to be consistent with the ΛCDM-allowed DM halo masses. The dark matter component is thus assumed to be pressureless dust. The calculations furthermore impose the condition that the LMC and SMC had an encounter within 20 kpc of each other between 1 and 4 Gyr ago in order to allow the Magellanic Stream to form, and they take into account cosmological expansion to constrain the relative positions of the galaxies 5 and more Gyr ago. To check for the consistency of the solutions, two independent statistical methods were applied to search for all allowed initial conditions within the error bars for the transverse velocity components of the LMC and SMC. The two tests yielded indistinguishable results. The calculations based on rigid Navarro–Frenk–White (NFW) DM halo profiles are conservative because equivalent simulations with live DM halos lead to faster merging [35]. These calculations show there to be no orbital solutions and that the observed system comprising the MW/LMC/SMC plus Magellanic Stream cannot be understood in the presence of DM halos in the context of the ΛCDM model. The problem that gravitational theory leads to the observed system is left for a future contribution.

## 2. The Model

### 2.1. NFW Profiles

When testing a theory for consistency with data, it is of paramount importance to not mix the two: purely theoretically calculated properties need to be compared with empirical data that have not been modulated by the very theory to be tested, and vice versa. It is therefore not permissible to choose sub-massive DM halos that allow a solution in order to argue that $\Lambda$CDM or $\Lambda$WDM theory accounts for the MW/LMC/SMC system. Thus, here, we are adamant at requiring the use of the theoretically predicted NFW dark matter halo profiles [43]. These arise from self-consistent cosmological structure formation simulations such as those documented in [44] for the range of baryonic galaxy masses considered here. More generally, the NFW DM halo profile is a standardised outcome of structure formation simulations in $\Lambda$CDM theory (e.g., [45]), and cold or warm DM halos have been shown to have similar overall profiles and densities [6–9]. While the inner and outer DM halo profiles can differ from the NFW form, the latter well represents the distribution of theoretically predicted DM particles around the baryonic component of a galaxy, especially for galaxies that are on a first-infall orbit (e.g., [46–49]). In seeking orbital solutions, the stellar masses are allowed to vary within $\pm 30$ per cent of their nominal values with the correspondingly different DM halo masses (Table 1 below). The range of theoretically predicted DM halo masses ($-41$ to $+68$ per cent for the MW, $-42$ to $+14$ per cent for the LMC and $-17$ to $+16$ per cent for the SMC) is thus accounted for.

The DM halo of either galaxy is treated as a rigid halo with a density profile according to [43] (NFW profile), truncated at the radius $R_{200}$:

$$\rho(r) = \frac{\rho_0}{r/R_s(1 + r/R_s)^2} \, , \tag{1}$$

where $R_s = R_{200}/c$, with $R_{200}$ denoting the radius yielding an average density of the halo of 200 times the cosmological critical density,

$$\rho_{crit} = \frac{3H^2}{8\pi G} \, , \tag{2}$$

and the concentration parameter $c$ [50],

$$\log_{10} c = 1.02 - 0.109 \left( \log_{10} \frac{M_{\text{vir}}}{10^{12} M_\odot} \right) \, . \tag{3}$$

The DM halo masses are derived from the stellar masses of the galaxies by means of Figure 7 of [44]. Note that using rigid DM halo profiles in such orbital computations is admissible because they have been verified to give conservative solutions (slower merging times) than self-consistent simulations with live DM halos [35] and, in particular, because the MW/LMC/SMC system has only a recent encounter history.

### 2.2. Dynamical Friction and Equations of Motion

Exploring the dynamics of bodies orbiting in the interior of DM halos implies that dynamical friction[1] has to be taken into account in an appropriate manner [51]. Here, the formulation of Chandrasekhar's dynamical friction as used in [35], their Equations (1) and (2), is applied.

Chandrasekhar's formula gives a quick-to-compute estimate for orbital decay, which is needed for the sake of establishing statistical statements about merger rates between galaxies. High-resolution simulations of live self-consistent systems confirm our approach of employing this semi-analytical formula in our three-body calculations (see Section 7 as well as Figures 13 and 14 in [35]). Using computationally significantly more time-intensive live simulations, in fact, leads to more rapid orbital decay such that our here-employed semi-analytical approach is conservative by allowing a larger range of solutions than would be the case if live DM halos were to be used.

The equations of motion for the individual galaxies are as given in Appendix C of [35][2] and are here augmented with an additional term taking into account the Hubble flow as follows: Based on the assumption of a flat universe (curvature parameter $k = 0$), we extended the equations of motion by the cosmic acceleration term $\frac{\ddot{s}}{s}\vec{r}_i$ caused by the Hubble flow of the expanding universe. Here, $s(t)$ is the scale factor of the Universe, and $\vec{r}_i$ is the position of a galaxy in the centre-of-mass frame of the group. In Cartesian coordinates, the equations of motion are then given by ($k = 1, 2, 3$ for the MW, LMC and SMC, respectively)

$$\frac{d^2}{dt^2}(\vec{r}_i)_k = m_i \cdot \frac{\ddot{s}}{s}(\vec{r}_i)_k + (\vec{F}_i)_k , \tag{4}$$

with

$$(\vec{F}_i)_k = \sum_{j \neq i} \left[ -\frac{\partial}{\partial(r_i)_k} V_{ij} + (\vec{F}_{ij}^{DF})_k - (\vec{F}_{ji}^{DF})_k \right]. \tag{5}$$

Here, $\vec{F}_i$ is the total force acting on galaxy i, $V_{ij}$ is the potential energy between galaxies i and j, and $\vec{F}_{ij}^{DF}$ is the dynamical friction force acting on galaxy i caused by the overlap of the DM halos of galaxies i and j (according to actio est reactio, $\vec{F}_{ji}^{DF}$ is taken into account, too).

Assuming a flat dark energy-dominated cosmology ($\Omega_{m,0} + \Omega_{\Lambda,0} = 1$), the second Friedmann equation becomes

$$\frac{\ddot{s}}{s} = H_0^2 \left( \Omega_{\Lambda,0} - \frac{1}{2}\Omega_{m,0} s^{-3} \right), \tag{6}$$

where $\Omega_{m,0}$ and $\Omega_{\Lambda,0}$ are the matter and dark energy densities at the present time, respectively, scaled by $\rho_{crit}$, and $H_0$ is the Hubble constant. Setting the conditions $s(0) = 0$ and $\dot{s} = H_0$ at $s(t) = 1$ (present time) yields an analytical expression for the cosmic-scale factor:

$$s(t) = \left(\frac{\Omega_{m,0}}{\Omega_{\Lambda,0}}\right)^{\frac{1}{3}} \sinh^{\frac{2}{3}}\left(\frac{3}{2}\sqrt{\Omega_{\Lambda,0}}H_0 t\right), \tag{7}$$

where $t$ is the age of the Universe. For our calculations, we used the following values: $\Omega_{m,0} = 0.315$, $\Omega_{\Lambda,0} = 0.685$ and $H_0 = 67.3$ km/s/Mpc. However, the cosmic acceleration term does not play a significant role for the time frame considered here. The order of magnitude contribution, compared to the forces between the DM halos in Equation (4), is in the range from $10^{-3}$ to $10^{-2}$ within the time range $[-5$ Gyr, today$]$. All numerical computations were performed using SAP's ABAP development workbench.

## 3. Observational Data

For the sake of easy accessibility, we list the observational constraints in Tables 1–7, and the references regarding the basic observational data in Table 2.

As mentioned in Section 2.1, we derived the DM halo masses of the MW, LMC and SMC according to [44] based on the stellar masses extracted from the references given in Table 2. Our approach, therefore, is in full correspondence with the predictions of ΛCDM regarding the DM halo masses. However, in order to overcome possible uncertainties in that regard and to achieve more independent results, we decided to apply our statistical evaluations in Section 4 individually to 27 mass combinations by varying the stellar mass of each galaxy by plus/minus 30%, as displayed in Table 1. The 27 mass combinations are then denoted by the indicators "o" for the original masses, "m" for −30% and "p" for +30%, for instance, "o-m-p" for the MW/LMC/SMC mass triple.

**Table 1.** Stellar masses of the galaxies (model (o)), varied by $-30\%$ (model (m)) and $+30\%$ (model (p)), and the derived DM halo masses according to Section 2.1.

| Object | Model | Stellar Mass $[M_\odot]$ | DM Halo Mass $[M_\odot]$ |
|--------|-------|--------------------------|--------------------------|
| MW | (o) | $5 \times 10^{10}$ | $2.41 \times 10^{12}$ |
|  | $-30\%$ (m) | $3.5 \times 10^{10}$ | $1.39 \times 10^{12}$ |
|  | $+30\%$ (p) | $6.5 \times 10^{10}$ | $4.05 \times 10^{12}$ |
| LMC | (o) | $3.2 \times 10^{9}$ | $2.55 \times 10^{11}$ |
|  | $-30\%$ (m) | $2.24 \times 10^{9}$ | $1.47 \times 10^{11}$ |
|  | $+30\%$ (p) | $4.16 \times 10^{9}$ | $2.90 \times 10^{11}$ |
| SMC | (o) | $5.3 \times 10^{8}$ | $1.07 \times 10^{11}$ |
|  | $-30\%$ (m) | $3.71 \times 10^{8}$ | $8.86 \times 10^{10}$ |
|  | $+30\%$ (p) | $6.89 \times 10^{8}$ | $1.24 \times 10^{11}$ |

**Table 2.** List of references for basic observational data.

| | |
|--|--|
| RA and DEC for LMC, SMC: | NASA Extragalactic Database |
| Distance LMC: | [52] |
| Distance SMC: | [53] |
| Radial velocities LMC, SMC: | [54] |
| Transverse velocities LMC: | [55] |
| Transverse velocities SMC: | [56] |
| Stellar mass MW: | [57] |
| Stellar mass LMC: | [58] |
| Stellar mass SMC: | [59] |
| Distance Galactic centre: | [60] |
| Solar circular speed: | [60] |
| Solar proper motion: | [61] |

The relevant coordinates and velocities are presented in Tables 3–5, and the transformation to the Cartesian heliocentric equatorial system is shown in Tables 6 and 7.

**Table 3.** Observational data for LMC and SMC and in parts for the Galactic centre.

| Object | RA (EquJ2000) | DEC (EquJ2000) | Heliocentric Distance | Heliocentric Radial Velocity |
|--------|---------------|----------------|-----------------------|------------------------------|
| LMC | $80.894°$ | $-69.756°$ | $49.97$ kpc | $262.2$ km/s |
| SMC | $13.187°$ | $-72.829°$ | $60.6$ kpc | $145.6$ km/s |
| MW | $266.405°$ | $-28.936°$ | $8.122$ kpc | |

**Table 4.** Transverse velocity components for LMC and SMC.

| Object | $v_{RA}$ [mas/yr] | $v_{RA}$ [km/s] | $v_{DEC}$ [mas/yr] | $v_{DEC}$ [km/s] |
|--------|-------------------|-----------------|--------------------|------------------|
| LMC | $1.872 \pm 0.045$ | $443.3 \pm 10.7$ | $0.224 \pm 0.054$ | $53.0 \pm 12.8$ |
| SMC | $0.820 \pm 0.060$ | $235.5 \pm 17.2$ | $-1.230 \pm 0.070$ | $-353.3 \pm 20.1$ |

**Table 5.** Circular velocity and proper motion of the Sun (Galactic coordinates).

| $v_c$ | $v_u$ | $v_v$ | $v_w$ |
|-------|-------|-------|-------|
| $233.34$ km/s | $11.10$ km/s | $12.24$ km/s | $7.25$ km/s |

**Table 6.** Cartesian equatorial coordinates and heliocentric equatorial velocities for the Magellanic Clouds and the centre of the Galaxy.

| Object | $x$ [kpc] | $y$ [kpc] | $z$ [kpc] | $v_x$ [km/s] | $v_y$ [km/s] | $v_z$ [km/s] |
|--------|-----------|-----------|-----------|--------------|--------------|--------------|
| LMC | 2.736 | 17.073 | −46.883 | −415.6 | 208.4 | −227.8 |
| SMC | 17.419 | 4.081 | −57.899 | −340.5 | 162.1 | −243.4 |
| MW | −0.446 | −7.094 | −3.930 | −114.4 | 120.4 | −181.4 |

**Table 7.** Transformation of the error bars in RA and DEC for the transverse velocity components of the Magellanic Clouds to Cartesian equatorial coordinates.

| Velocity Component | $\Delta v_x$ [km/s] | $\Delta v_y$ [km/s] | $\Delta v_z$ [km/s] |
|--------------------|---------------------|---------------------|---------------------|
| LMC RA | ±10.5 | ±1.69 | ±0 |
| LMC DEC | ±1.90 | ±11.9 | ±4.43 |
| SMC RA | ±3.93 | ±16.8 | ±0 |
| SMC DEC | ±18.7 | ±4.39 | ±5.95 |

The Magellanic Stream consists of $H_I$ gas, which trails behind both the LMC and the SMC across a large fraction of the sky. It is thought to be the result of a combination of tidal forces and ram pressure stripping through the orbit of the LMC and SMC within the hot gaseous halo of the MW. Studies of the origin of the Magellanic Stream have shown that it was most likely created when the LMC and the SMC had a close encounter, during which some of the gas of the LMC and SMC became less bound to be subsequently removed from the pair through ram pressure stripping [39,40,62,63].

## 4. Statistical Methods

In order to cross-check the solutions and to enhance the confidence in them, we utilised two independent statistical methods, a Markov-Chain Monte-Carlo method (MCMC, see Section 4.1) and the genetic algorithm (GA, see Section 4.2). The reason why we do so is given in the introductory statement of Section 4.2.

This is the initial situation for each of the MCMC and GA methods: due to the error bars of the transverse velocity components in RA and DEC for the Magellanic Clouds (see Table 4), we have to consider four open parameters $P_i$ regarding the calculation of the three-body orbits of the dark matter halos, as displayed in Table 8.

**Table 8.** Open parameters for the statistical methods and the corresponding $1\sigma$ uncertainty values from the observational data.

| $P_1$ (LMC: $v_{RA}$) | $P_2$ (LMC: $v_{DEC}$) | $P_3$ (SMC: $v_{RA}$) | $P_4$ (SMC: $v_{DEC}$) |
|-----------------------|------------------------|-----------------------|------------------------|
| $v_1$ | $v_2$ | $v_3$ | $v_4$ |
| 443.3 km/s | 53.0 km/s | 235.5 km/s | −353.3 km/s |
| $\sigma_1$ | $\sigma_2$ | $\sigma_3$ | $\sigma_4$ |
| 10.7 km/s | 12.8 km/s | 17.2 km/s | 20.1 km/s |

This means that by employing the MCMC method and the GA method separately for each of the 27 mass combinations (see Section 3), we search for solutions of the three-galaxy orbits with best fits to the transverse velocity components of the Magellanic Clouds. It is important to note that our goal is to achieve results independent of the particular masses of the galaxies, not to find a best-fit mass combination.

Further, based on the radio-astronomical observational data concerning the Magellanic $H_I$-Stream explained in Section 3, we specify the following broad condition, which

we hereinafter refer to as the condition COND, regarding admissible past orbits of the Magellanic Clouds: the *LMC and SMC encountered each other within the past time interval of [−4 Gyr, −1 Gyr] at a pericentre distance of less than* 20 kpc. Incorporating this condition, the algorithms searched for solutions by integrating Equation (4) backwards in time up to −5 Gyr.

### 4.1. Markov-Chain Monte Carlo (MCMC)

We followed a methodology proposed by [64] employing an affine-invariant ensemble sampler for the Markov-Chain Monte-Carlo method. A detailed guideline for implementation can be found in [65]. The basics of applying this formalism to our situation are outlined in Appendix D of [35].

#### 4.1.1. Definition of the Posterior Probability Density

First of all, concerning the open parameters, we need to account for the error bars of the transverse velocity components $v_i$ of the Magellanic Clouds. This is ensured by an appropriate definition of the prior distribution $p(\mathbf{X})$, where $\mathbf{X}$ symbolises the parameter vector $(P_1 \ldots P_4)$. With the values from Table 8, the four contributions to the prior distribution read ($i = 1, \ldots, 4$)

$$p(\mathbf{X})_i \propto \exp\left(-\frac{(P_i - v_i)^2}{2\sigma_i^2}\right) . \tag{8}$$

Exploiting the minimal distance $d_{23}$ between the LMC and SMC within the time period $[−4 \, \text{Gyr}, −1 \, \text{Gyr}]$, the condition COND implies the likelihood function

$$P_C(\mathbf{X}) \propto \begin{cases} 1, & d_{23} \leq d_{per} , \\ \exp\left(-\dfrac{(d_{23} - d_{per})^2}{2 \cdot d_0^2}\right), & d_{23} > d_{per} , \end{cases} \tag{9}$$

with $d_{per} = 20 \, \text{kpc}$ and $d_0 = 5 \, \text{kpc}$. The posterior probability density is then given by the product of Equations (8) and (9):

$$\pi(\mathbf{X}) \propto \prod_{i=1}^{4} p(\mathbf{X})_i \cdot P_C(\mathbf{X}) . \tag{10}$$

Here, the normalising constant for an overall probability of 1 is neglected because it is clear from the comparative search algorithm that the absolute values of $\pi(\mathbf{X})$ are irrelevant.

#### 4.1.2. The First Ensemble

Generating random integer numbers $k \in \{0, \ldots, 1000\}$ separately for each walker and for each open parameter of a given walker, the first ensembles are created according to ($i \in \{1, \ldots, 4\}$):

$$P_i = (v_i - n\sigma_i) + k \cdot 2n\sigma_i/1000 , \tag{11}$$

with varying $n$ to account for extended error bars, as explained in Section 4.3.

### 4.2. Genetic Algorithm (GA)

The general aspects of the genetic algorithm are explained in detail in [66]. As pointed out there, the major advantage of this method is perceived to be its capability of avoiding local maxima in the process of searching for the global maximum of a given function (here, the fitness function). This feature makes it worthwhile to employ the GA method as a second independent statistical approach besides the MCMC method.

A precise description of the algorithm, especially instructions for its implementation, can also be found in [67]. To put it in a nutshell, the open parameters are related to genes, which are concatenated to genotypes. The set of a number of genotypes is considered to be a population, where the members pairs produce a follow-up generation with new features due to randomly performed cross-over and mutation. Winners per generation are

determined by means of the fitness function, and by comparing the generations, the overall winner is found.

Comparing it to the more familiar MCMC method, we have the following parallels: walker <-> genotype, ensemble <-> population, set of ensembles <-> generations, posterior probability density <-> fitness function.

### 4.2.1. The GA Generations

Each open parameter from Table 8 is mapped to a 4-digit string ("gene") $[abcd]_i$ ($i \in \{1, \ldots, 4\}$),

$$P_i = (v_i - n\sigma_i) + [abcd]_i \cdot 2n\sigma_i / 10\,000 \,, \tag{12}$$

again with varying $n$, like in Section 4.1.2, in order to account for extended error bars, as explained in Section 4.3. All genes together define the genotypes as $4 \times 4$-digit strings,

$$\boxed{[abcd]_1 \ldots [abcd]_4} \,, \tag{13}$$

which generate a population of $N_{pop}$ genotypes. The first generation is established by randomly creating $N_{pop}$ $4 \times 4$-digit strings.

### 4.2.2. The Definition of the Fitness Function

Our goal is to establish GA evaluations that are compatible with the evaluations by means of the Markov-Chain Monte-Carlo method. Therefore, we chose the definition of the fitness function to be identical to the definition of the posterior probability density (Equation (10)),

$$\mathcal{F}(\mathbf{X}) = \prod_{i=1}^{4} p(\mathbf{X})_i \cdot P_C(\mathbf{X}) \,, \tag{14}$$

where the individual components are taken from Equations (8) and (9).

### 4.3. Results

Both statistical methods, MCMC and GA, deliver mutually consistent, and in fact indistinguishable, results.

### 4.3.1. Step I

As a first step, we tried to find solutions for which all four open parameters $P_i$ lie within the $1\sigma$ error bars of the transverse velocity components of the Magellanic Clouds (see Table 8). Employing the methods MCMC and GA as search engines, using 1000 ensembles with 100 walkers (MCMC) and, accordingly, 1000 generations with a population of 100 genotypes (GA), we repeated the search 100 times for both statistical methods for each of the 27 mass combinations according to Table 1. This attempt to find a solution failed. That is, within the here-probed $1\sigma$ uncertainty range of the transverse velocity components, neither algorithm found orbits that fulfil the condition COND. In other words, the MW/LMC/SMC plus Magellanic Stream system cannot exist in its observed configuration in the presence of dark matter halos.

To proceed further, we gradually extended the allowed intervals for the parameters $P_i$ to be $v_i \pm n\sigma_i$ with increasing integers $n$; see Equations (11) and (12). Neither the MCMC nor the GA statistical method found a solution for $\sigma = 1, 2, 3$. For $\sigma = 4$, both methods delivered solutions for a single mass combination only, namely, p-m-p. The details regarding all 27 mass combinations are given in Appendix A.

### 4.3.2. Step II

How to go about quantifying the first results obtained in Section 4.3.1? First of all, for a mass combination with the first solutions based on $n\sigma$ error intervals, we chose the $(n + 1)\sigma$ intervals to be allowed for the open parameters. This is motivated by the thought

that relaxing the overall $n\sigma$ condition for all open parameters may result in solutions with individual smaller-than-$1\sigma$ deviations of the parameters.

Furthermore, we constructed a "probability-sigma-grid" in the following sense: If a parameter $P_i$ lies within the error bar ($< 1\sigma_i$), then its probability weighting factor is 1, a conservative cautious setting. If a parameter $P_i$ lies outside the error bar, then its deviation from $v_i$ is weighted with the remainder of the probability function based on $0.1\sigma$ intervals. For instance, if we have $P_i = v_i + 3.74\sigma_i$, then the corresponding probability weighting factor for this parameter is calculated according to the remainder of the probability function for $3.7\sigma$.

Based on the same search method as in Section 4.3.1, we searched for best-fit solutions by calculating an overall probability for the combination of the open parameters. The results are displayed in Tables 9 and 10 for the methods MCMC and GA, respectively.

**Table 9.** *Results using the MCMC method*: Probabilities regarding the evaluation of the error intervals of the plane-of-sky velocity components of the LMC and SMC for the individual best-fit solutions of each mass combination considered, as explained in Section 4.3.2.

| Combination of Masses | Deviation of $P_1$ | Deviation of $P_2$ | B of $P_3$ | Deviation of $P_4$ | Probability |
|---|---|---|---|---|---|
| o-o-o | >6.8$\sigma$ | >3.9$\sigma$ | <1$\sigma$ | >3.6$\sigma$ | $3.2 \times 10^{-19}$ |
| o-o-m | >6.3$\sigma$ | >3.1$\sigma$ | <1$\sigma$ | >1.7$\sigma$ | $5.1 \times 10^{-14}$ |
| o-o-p | >6.8$\sigma$ | <1$\sigma$ | <1$\sigma$ | >6.8$\sigma$ | $1.1 \times 10^{-22}$ |
| o-m-o | >5.0$\sigma$ | <1$\sigma$ | <1$\sigma$ | >5.7$\sigma$ | $6.9 \times 10^{-15}$ |
| o-m-m | >3.4$\sigma$ | >1.0$\sigma$ | <1$\sigma$ | >6.8$\sigma$ | $7.1 \times 10^{-15}$ |
| o-m-p | >4.3$\sigma$ | >1.2$\sigma$ | >1.1$\sigma$ | >5.9$\sigma$ | $3.9 \times 10^{-15}$ |
| o-p-o | >6.5$\sigma$ | >2.4$\sigma$ | >1.4$\sigma$ | >4.3$\sigma$ | $3.6 \times 10^{-18}$ |
| o-p-m | >5.9$\sigma$ | >2.3$\sigma$ | <1$\sigma$ | >2.1$\sigma$ | $2.8 \times 10^{-12}$ |
| o-p-p | >6.7$\sigma$ | >1.8$\sigma$ | >4.3$\sigma$ | >6.9$\sigma$ | $1.3 \times 10^{-28}$ |
| m-o-o | >5.7$\sigma$ | >1.2$\sigma$ | >3.2$\sigma$ | >6.9$\sigma$ | $2.0 \times 10^{-23}$ |
| m-o-m | >6.8$\sigma$ | >1.0$\sigma$ | <1$\sigma$ | >6.9$\sigma$ | $5.5 \times 10^{-23}$ |
| m-o-p | >6.2$\sigma$ | >2.4$\sigma$ | <1$\sigma$ | >6.9$\sigma$ | $4.8 \times 10^{-23}$ |
| m-m-o | >5.1$\sigma$ | <1$\sigma$ | <1$\sigma$ | >6.0$\sigma$ | $6.7 \times 10^{-16}$ |
| m-m-m | >5.1$\sigma$ | >1.0$\sigma$ | <1$\sigma$ | >6.3$\sigma$ | $1.0 \times 10^{-16}$ |
| m-m-p | >5.4$\sigma$ | <1$\sigma$ | >1.0$\sigma$ | >5.8$\sigma$ | $4.4 \times 10^{-16}$ |
| m-p-o | >6.9$\sigma$ | >1.5$\sigma$ | >2.8$\sigma$ | >6.9$\sigma$ | $1.9 \times 10^{-26}$ |
| m-p-m | >6.8$\sigma$ | <1$\sigma$ | >4.8$\sigma$ | >6.8$\sigma$ | $1.7 \times 10^{-28}$ |
| m-p-p | >6.9$\sigma$ | >2.8$\sigma$ | >1.1$\sigma$ | >6.8$\sigma$ | $7.6 \times 10^{-26}$ |
| p-o-o | >6.2$\sigma$ | <1$\sigma$ | <1$\sigma$ | >7.9$\sigma$ | $1.6 \times 10^{-24}$ |
| p-o-m | >6.5$\sigma$ | >1.9$\sigma$ | >2.1$\sigma$ | >7.9$\sigma$ | $4.6 \times 10^{-28}$ |
| p-o-p | >5.9$\sigma$ | >1.9$\sigma$ | <1$\sigma$ | >7.7$\sigma$ | $2.9 \times 10^{-24}$ |
| p-m-o | >4.9$\sigma$ | >1.4$\sigma$ | <1$\sigma$ | >5.8$\sigma$ | $1.0 \times 10^{-15}$ |
| p-m-m | >5.4$\sigma$ | <1$\sigma$ | <1$\sigma$ | >5.9$\sigma$ | $2.4 \times 10^{-16}$ |
| p-m-p | >4.7$\sigma$ | <1$\sigma$ | >2.0$\sigma$ | >3.5$\sigma$ | $5.5 \times 10^{-11}$ |
| p-p-o | >7.7$\sigma$ | >1.2$\sigma$ | <1$\sigma$ | >7.9$\sigma$ | $8.8 \times 10^{-30}$ |
| p-p-m | >7.3$\sigma$ | <1$\sigma$ | >1.4$\sigma$ | >8.4$\sigma$ | $2.1 \times 10^{-30}$ |
| p-p-p | >6.9$\sigma$ | >2.0$\sigma$ | >1.0$\sigma$ | >7.8$\sigma$ | $1.5 \times 10^{-27}$ |

Regarding the MCMC method, it is interesting to note that, in some cases, individual parameters are not confined to the mentioned $(n + 1)\sigma$ intervals because the stretch moves can place walkers outside of the intervals specified by Equation (11), thus supporting the overall process of maximising the posterior probability density consisting of all open parameters. This is not possible for the GA method, as the genotypes are a priori confined to the preset intervals.

**Table 10.** Same as for Table 9, but for the *GA method*.

| Combination of Masses | Deviation of $P_1$ | Deviation of $P_2$ | Deviation of $P_3$ | Deviation of $P_4$ | Probability |
|---|---|---|---|---|---|
| o-o-o | >6.8σ | >3.0σ | >1.1σ | >4.0σ | $4.9 \times 10^{-19}$ |
| o-o-m | >6.4σ | <1σ | <1σ | >3.1σ | $3.0 \times 10^{-13}$ |
| o-o-p | >6.5σ | >1.2σ | <1σ | >6.9σ | $9.7 \times 10^{-23}$ |
| o-m-o | >4.5σ | >1.6σ | <1σ | >5.9σ | $2.7 \times 10^{-15}$ |
| o-m-m | >4.7σ | >1.0σ | <1σ | >6.3σ | $7.8 \times 10^{-16}$ |
| o-m-p | >4.4σ | <1σ | >1.9σ | >5.9σ | $2.3 \times 10^{-15}$ |
| o-p-o | >6.4σ | >3.5σ | <1σ | >3.5σ | $3.4 \times 10^{-17}$ |
| o-p-m | >5.9σ | >3.0σ | <1σ | >1.9σ | $5.6 \times 10^{-13}$ |
| o-p-p | >5.9σ | <1σ | >6.0σ | >6.9σ | $3.8 \times 10^{-29}$ |
| m-o-o | >6.3σ | >1.5σ | >1.4σ | >6.9σ | $3.4 \times 10^{-23}$ |
| m-o-m | >6.9σ | >1.5σ | >1.6σ | >6.8σ | $8.0 \times 10^{-25}$ |
| m-o-p | >6.8σ | >1.0σ | <1σ | >6.7σ | $2.2 \times 10^{-22}$ |
| m-m-o | >4.6σ | >1.7σ | >1.0σ | >6.1σ | $4.0 \times 10^{-16}$ |
| m-m-m | >5.2σ | <1σ | >2.1σ | >5.8σ | $4.7 \times 10^{-17}$ |
| m-m-p | >5.3σ | >1.0σ | >1.4σ | >5.6σ | $4.0 \times 10^{-16}$ |
| m-p-o | >6.9σ | >1.4σ | >3.5σ | >6.8σ | $4.1 \times 10^{-27}$ |
| m-p-m | >6.9σ | >1.3σ | >4.0σ | >6.9σ | $3.3 \times 10^{-28}$ |
| m-p-p | >6.8σ | <1σ | >3.0σ | >6.9σ | $1.5 \times 10^{-25}$ |
| p-o-o | >6.9σ | >1.9σ | >6.0σ | >6.9σ | $3.1 \times 10^{-33}$ |
| p-o-m | >7.3σ | <1σ | >1.1σ | >7.8σ | $4.9 \times 10^{-28}$ |
| p-o-p | >6.3σ | >1.0σ | >4.9σ | >6.9σ | $1.5 \times 10^{-27}$ |
| p-m-o | >4.9σ | <1σ | >1.2σ | >5.8σ | $1.5 \times 10^{-15}$ |
| p-m-m | >3.4σ | >2.0σ | <1σ | >6.9σ | $1.6 \times 10^{-16}$ |
| p-m-p | >5.5σ | >1.0σ | >1.2σ | >2.7σ | $6.1 \times 10^{-11}$ |
| p-p-o | >7.5σ | <1σ | >2.9σ | >7.9σ | $6.7 \times 10^{-31}$ |
| p-p-m | >7.8σ | >4.1σ | >4.6σ | >7.7σ | $1.5 \times 10^{-38}$ |
| p-p-p | >7.4σ | >1.9σ | <1σ | >7.5σ | $5.0 \times 10^{-28}$ |

The result in Section 4.3.1 that the mass combination p-m-p provides the best-fit solution is confirmed by either statistical method, MCMC and GA. Furthermore, Figure 1 shows the trajectories, displayed as pairwise distances, of the individual best-fit solutions for the mass combinations o-o-o and p-m-p. Even for that detail, **both methods** deliver identical results.

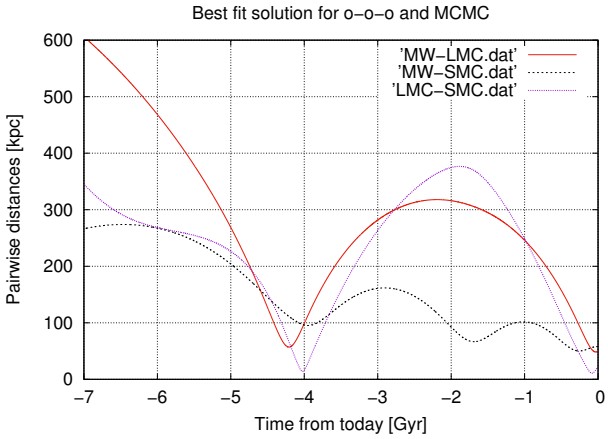
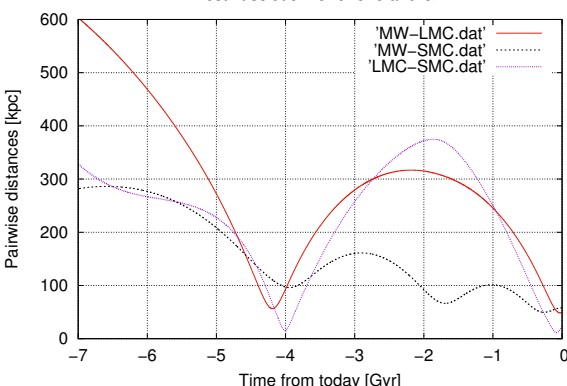

**Figure 1.** *Cont.*

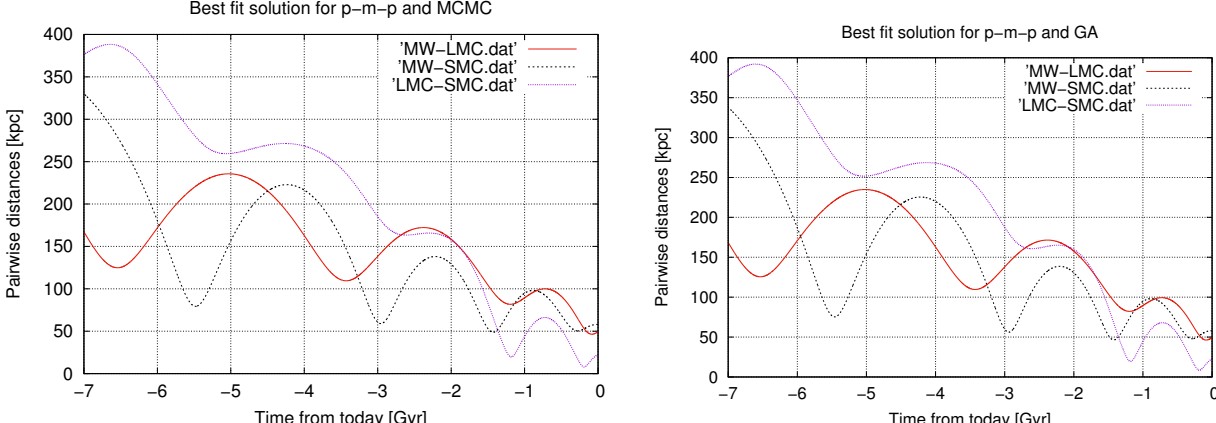

**Figure 1.** Orbits, displayed as pairwise distances, calculated backwards in time up to $-7\,\mathrm{Gyr}$ for the best-fit solutions for the mass combinations o-o-o and p-m-p with either statistical method, MCMC or GA. **Left** panel: Orbits obtained using the MCMC method. **Right** panel: Orbits obtained using the GA method.

### 4.3.3. Step III

Especially due to the MCMC method not being utilised as a pure search engine only, we established its validity by checking the results obtained in the previous Section 4.3.2 in terms of the autocorrelation function for the mass combinations o-o-o (original masses) and p-m-p (mass combination with the overall best-fit solutions according to Sections 4.3.1 and 4.3.2) and creating sets of 1000 solutions in each case [3]. For the GA method, we undertook broader search runs, establishing 1000 solutions for the mentioned mass combinations.

The autocorrelation functions are presented in Figures 2 and 3, demonstrating that convergence is achieved quickly for the MCMC method, and the improved probabilities are displayed in Tables 11 and 12 for the MCMC and GA methods, respectively.

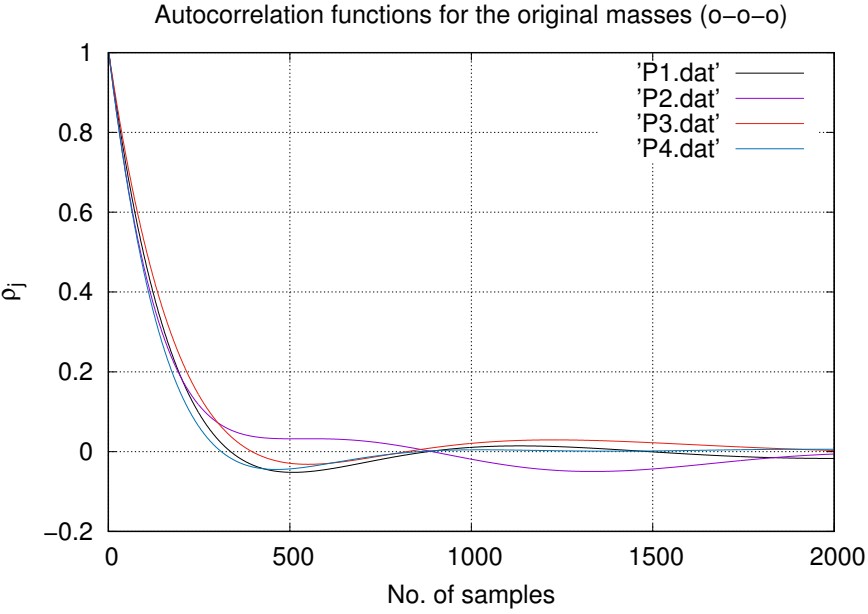

**Figure 2.** *The MCMC method*: Autocorrelation functions related to the open parameters for the primordial mass combination o-o-o. The calculation is based on a set of 50,000 ensembles.

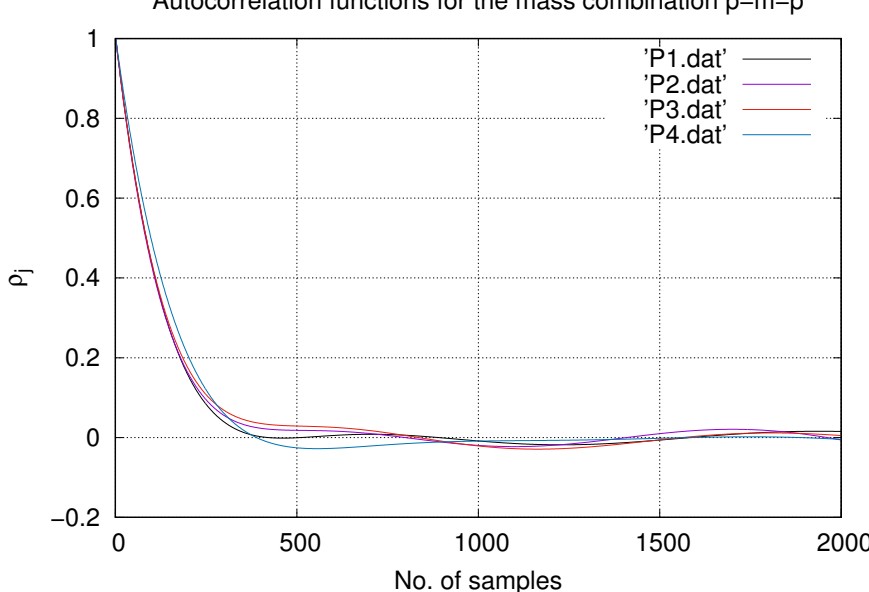

**Figure 3.** *The MCMC method*: Same as for Figure 2 but for the overall best-fit mass combination p-m-p.

**Table 11.** *Results from the MCMC method*: Probabilities regarding the evaluation of the error intervals of the plane-of-sky velocity components of the LMC and SMC for the individual best-fit solutions of the mass combinations o-o-o and p-m-p based on sets of 1000 solutions, as explained in Section 4.3.3.

| Combination of Masses | Deviation of $P_1$ | Deviation of $P_2$ | Deviation of $P_3$ | Deviation of $P_4$ | Probability |
|---|---|---|---|---|---|
| o-o-o | $> 6.8\sigma$ | $> 3.9\sigma$ | $< 1\sigma$ | $> 3.6\sigma$ | $3.2 \times 10^{-19}$ |
| p-m-p | $> 5.3\sigma$ | $> 2.8\sigma$ | $> 1.0\sigma$ | $> 1.0\sigma$ | $5.9 \times 10^{-10}$ |

**Table 12.** Same as for Table 11, but for the *GA method*.

| Combination of Masses | Deviation of $P_1$ | Deviation of $P_2$ | Deviation of $P_3$ | Deviation of $P_4$ | Probability |
|---|---|---|---|---|---|
| o-o-o | $> 6.9\sigma$ | $> 3.2\sigma$ | $< 1\sigma$ | $> 3.6\sigma$ | $2.3 \times 10^{-18}$ |
| p-m-p | $> 5.5\sigma$ | $> 2.2\sigma$ | $< 1\sigma$ | $> 1.6\sigma$ | $1.2 \times 10^{-10}$ |

### 4.3.4. Interpretation

In Step I (Section 4.3.1), it was shown that no orbital solutions are possible if the observational quantities are allowed to span the $\pm 1\,\sigma$ uncertainty range. In Step II (Section 4.3.2), the uncertainty range was therefore increased to $\pm(n+1)\,\sigma$, with the result that values of $n$ are needed to obtain viable orbital solutions that lie outside the $5\,\sigma$ confidence range of the quantities such that the orbital solutions are likely with probabilities smaller than $10^{-9}$ (Tables 9 and 10). Step III (Section 4.3.3) demonstrates that the MCMC and GA methods yield indistinguishable results, thus confirming the conclusion that orbital solutions do not exist for the observed MW/LMC/SMC plus Magellanic Stream system in the presence of the theoretically expected DM halos.

The dynamical behaviour of the MW/LMC/SMC system demonstrates the importance of dynamical friction in the context of interacting galaxies, as dynamical friction significantly influences the orbits. Figure 4 shows that the forces due to dynamical friction between the LMC and the SMC are comparable to the pure gravitational force between the overlapping DM halos.

As a thought experiment, we calculated the orbits back in time to $-7$ Gyr *with* and *without* dynamical friction for the overall best-fit solution derived for the mass combination

p-m-p and for the best-fit solution for the original mass combination o-o-o. Dynamical friction can be turned off by omitting the terms $\vec{F}^{DF}$ in Equation (5). This retains the gravitational pull of the DM halo but avoids the deceleration through Chandrasekhar dynamical friction. This Gedanken experiment is thus a rough approximation of the situation in Milgromian dynamics, according to which a galaxy generates a Milgromian gravitational potential that can be viewed as stemming from a Newtonian plus phantom DM halo that does not generate dynamical friction. With this approximation of MOND, the GA method as the search engine immediately delivers 17 mass combinations already within the $1\sigma$ intervals (see also Appendix A). In other words, solutions matching the error intervals for the transverse velocities of the LMC and SMC are found readily without Chandrasekhar's dynamical friction but with the potential generated by the DM halo. The result is displayed in Figure 5. With the absence of Chandrasekhar dynamical friction on DM halos, the Magellanic Clouds would have a long orbital lifetime as satellites of the MW, possibly being massive tidal dwarf galaxies formed during the MW–Andromeda encounter about 10 Gyr ago.

These results thus suggest that the MW/LMC/SMC plus Magellanic Stream system may have a straightforward orbital solution in Milgromian dynamics, and it is thus of much interest to simulate this system and its possible origin in this non-Newtonian framework.

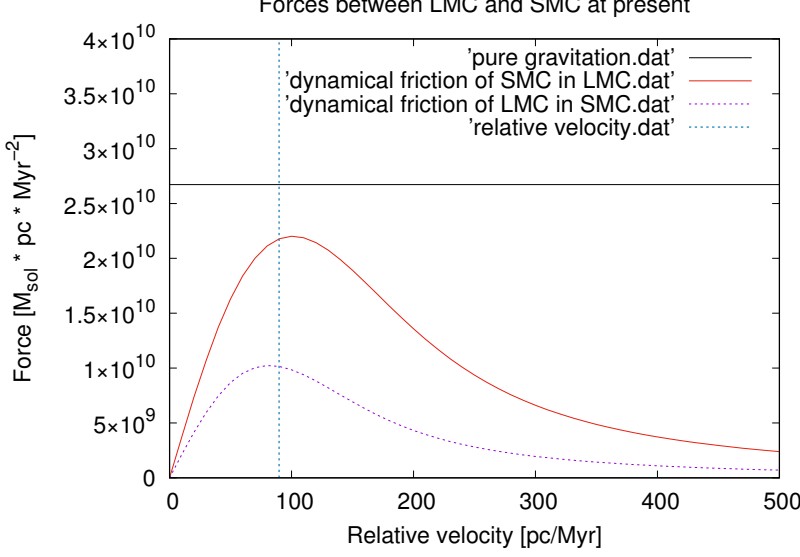

**Figure 4.** Forces between the Magellanic Clouds based on today's observational data with a distance of 22.5 kpc between the CMs of the dark matter halos and the present-day relative velocity of 89.8 pc/Myr (87.8 km/s), shown as the vertical dotted line.

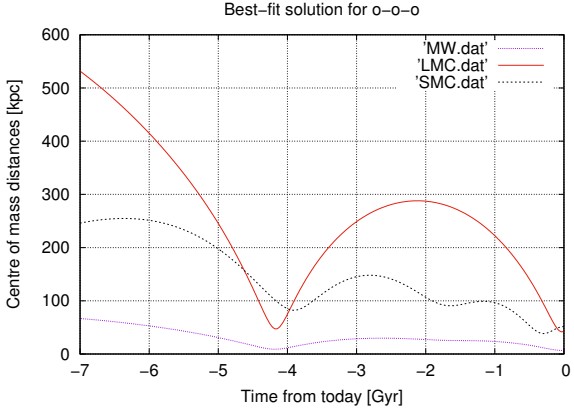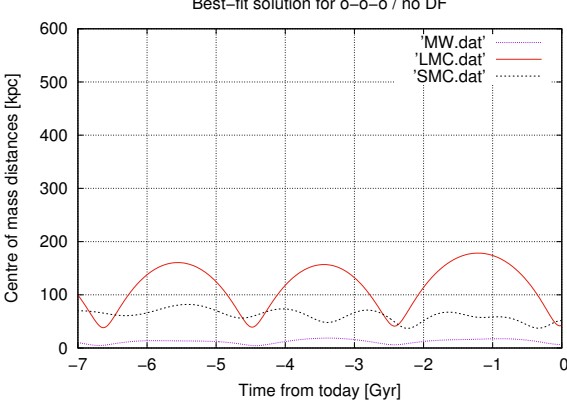

**Figure 5.** *Cont.*

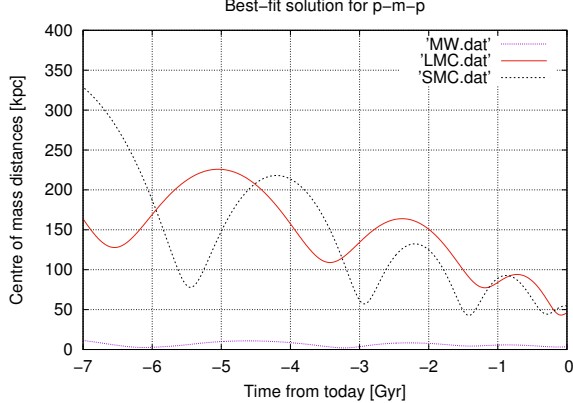
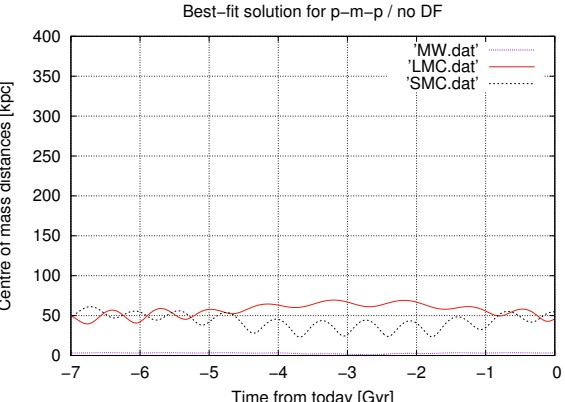

**Figure 5.** Centre-of-mass distances, calculated backwards in time up to −7 Gyr for the best-fit solutions for the mass combinations o-o-o and p-m-p. Note that the best-fit solutions delivered by the MCMC and GA methods are indistinguishable (see Figure 1). **Left** panel: Full calculation including dynamical friction. **Right** panel: Based on identical initial conditions at present, orbits obtained by switching off dynamical friction.

## 5. Conclusions

The conclusions reached by this analysis are very robust, since both the MCMC and GA algorithms lead to indistinguishable results. Taking the observed configuration in the six-dimensional phase space of the MW/LMC/SMC plus Magellanic Stream system as the necessary boundary condition, it is impossible to find orbital solutions backwards in time that fulfil the very liberal condition COND (Section 4). The orbits accelerate too rapidly (backwards in time) such that the LMC and SMC could not have remained bound long enough to have the required encounter that is needed to have occurred to produce the Magellanic Stream. In other words, the system merges too rapidly forward in time to allow a close encounter as defined through COND and to still be visible today as two distinctly separated galaxies next to the MW. Solutions do not even appear if the DM halo masses of the three galaxies are allowed to be up to 42 per cent larger or smaller than those under the standard assumptions (see Table 1). The possibility that the LMC and SMC fell into the MW DM halo independently of each other but at a similar time in order to allow them to pair up to the observed binary is arbitrarily unlikely because the LMC and SMC would have to have had relative velocities to each other and to the MW that oppose the Hubble flow. Such an unlikely solution is, in any case, not possible because the condition COND requires the LMC and SMC to have had a close encounter at a time in the past such that the two would have merged today due to the mutual dynamical friction on their respective DM halos. In any case, neither the MCMC nor the GA method found such solutions. This work thus shows that the observed configuration of the MW/LMC/SMC plus Magellanic Stream system is not possible in the presence of the theoretically expected DM halos.

The results based on the Chandrasekhar dynamical friction test applied to the MW/-LMC/SMC triple system arrived at here corroborate the previous evidence based on the same test but other systems, noted in the Introduction, that questions the existence of dark matter particles. The independently documented problems [5,68–74] of fitting the standard model of cosmology to the observed Universe on most probed scales are consistent with these results.

The thought experiment in Section 4.3.4, in which the potentials of the DM halos are kept but the Chandrasekhar dynamical friction term is set to zero, naturally leads to solutions. This experiment is an approximation to the situation in Milgromian dynamics and demonstrates that the origin and evolution of the MW/LMC/SMC plus Magellanic Stream system need to be studied in this non-Newtonian framework.

**Author Contributions:** Conceptualisation, W.O. and P.K.; methodology, W.O. and P.K.; software, W.O.; validation, W.O. and P.K.; formal analysis, W.O.; resources, W.O.; data curation, W.O.; writing—original draft preparation, W.O. and P.K.; writing—review and editing, W.O. and P.K.; visualisation, W.O. All authors have read and agreed to the published version of the manuscript.

**Funding:** This research received no external funding.

**Data Availability Statement:** All data underlying this research are available in this article (see Section 3).

**Acknowledgments:** W.O. acknowledges the support of scdsoft AG in providing an SAP system environment for the numerical calculations. Without the support of scdsoft's executives *P. Pfeifer* and *U. Temmer*, the innovative approach of programming the numerical tasks in SAP's language ABAP would not have been possible.

**Conflicts of Interest:** Wolfgang Oehm was an employee of scdsoft AG. The authors declare that the research was conducted in the absence of any commercial or financial relationships that could be construed as a potential conflict of interest.

## Abbreviations

The following abbreviations are used in this manuscript:

| | |
|---|---|
| ABAP | SAP's programming language |
| CDM | Cold dark matter |
| COND | Condition specified in Section 4 |
| DEC | Declination |
| DM | Dark matter |
| GA | Genetic algorithm |
| ΛCDM | Dark-energy plus cold-dark-matter model of cosmology |
| ΛWDM | Dark-energy plus warm-dark-matter model of cosmology |
| LMC | Large Magellanic Cloud |
| MCMC | Markov-Chain Monte Carlo |
| MW | The Galaxy (Milky Way) |
| NED | NASA/IPAC extragalactic database |
| NFW | Navarro, Frenk and White profile |
| RA | Right ascension |
| SMC | Small Magellanic Cloud |

## Appendix A. First Search Results

Confining the solutions to $n\sigma$ intervals simultaneously for all transverse velocity components of the Magellanic Clouds, and based on 100 attempts for each $n\sigma$ interval to find an appropriate solution for each of the 27 mass combinations, we obtained the following results for the MCMC method:

$1\sigma$ : none;

$2\sigma$ : none;

$3\sigma$ : none;

$4\sigma$ : p-m-p;

$5\sigma$ : o-m-p, m-m-p, p-m-o;

$6\sigma$ : o-m-o, o-m-m, o-p-m, m-m-o, m-m-m, p-m-m;

$7\sigma$ : o-o-o, o-o-m, o-o-p, o-p-o, o-p-p, m-o-o, m-o-m, m-o-p, m-p-o, m-p-m, m-p-p, p-o-o;

$8\sigma$ : p-o-m, p-o-p, p-p-o, p-p-m, p-p-p.

For the GA method, we obtained the following:

$1\sigma$ : 　none;

$2\sigma$ : 　none;

$3\sigma$ : 　none;

$4\sigma$ : 　p-m-p;

$5\sigma$ : 　o-m-p, o-p-m, m-m-p, p-m-o;

$6\sigma$ : 　o-o-m, o-m-o, o-m-m, o-p-o, m-o-p, m-m-o,
　　　　　m-m-m, p-m-m;

$7\sigma$ : 　o-o-o, o-o-p, o-p-p, m-o-o, m-o-m, m-p-o, m-p-m,
　　　　　m-p-p, p-o-o, p-o-m, p-o-p, p-p-o, p-p-p;

$8\sigma$ : 　p-p-m.

However, when neglecting the effect of dynamical friction (by omitting the terms $\vec{F}^{DF}$ in Equation (5)), the GA method already delivers 17 mass combinations with appropriate solutions within the $1\sigma$ intervals:

o-m-p, m-o-o, m-o-m, m-o-p, m-m-o, m-m-m, m-m-p, m-p-o, m-p-m, m-p-p, p-o-o, p-o-m, p-o-p, p-m-p, p-p-o, p-p-m, p-p-p.

## Notes

[1]　The term "dynamical dissipation" appears to be the more appropriate than "dynamical friction" because the physical process is not friction but the distortion of the orbits of the individual particles of the dark matter halos due to the long-distance gravitational forces. However, as the term "dynamical friction" is established in the community, we retain this term.

[2]　There is a printing mistake in the equation for $\Phi_i(s_i)$, the third equation in Appendix C in [35]: the $-$ sign in the first row is incorrect.

[3]　As explained in [35] the autocorrelation functions deliver an estimate for the convergence of the search algorithm for sufficiently large numbers of ensembles. Therefore, in order to establish full convergence for the approximate calculation of the autocorrelation functions, we created 50,000 ensembles for both mass combinations considered, o-o-o and p-m-p. The mentioned 1000 solutions fulfilling the conditions $(n+1)\sigma$ intervals and COND were extracted from follow-up runs based on the set of walkers of ensemble 50,000 in either mass combination case.

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
