# Peer review of "The Relevance of Dynamical Friction for the MW/LMC/SMC Triple System"

_universe, doi:10.3390/universe10030143_

Round 1

Reviewer 1 Report

Comments and Suggestions for Authors

This is an important manuscript for reasons that I explain in this review. When Vera Rubin measured the rotation curves of external galaxies and deduced missing matter, the Dark Matter, she guessed that it resides in the halo of galaxies. This paper uses computational physics to test that assumption. They first show that the Milky Way and the Magellanic Clouds are negligibly perturbed by the expansion of the Universe, in co-moving coordinates. The computational physics is the solution of the 3-body Milky Way galaxy (MW), Large Magellanic Cloud (LMC) and Small Magellanic Cloud (SMC) trajectories computed backward by including dynamical friction of each Dark Matter halo postulated to exist. They require that the backward extrapolated trajectories have a requirement that the LMC and SMC had an encounter less than 20 kpc within the past 4 Gyr, to explain the data associated with the Magellanic hydrogen stream. They used state-of-the-art random algorithms to probe the possible phase space and deduce that the presence of Dark Matter halos prevent a solution.

This is a very good computational physics paper and addresses a neglected issue: dynamical friction associated with Dark Matter halos.

Had Rubin access to MW stellar rotation data, she would have discovered that the MW does not have a unique rotation curve and therefore her Dark Matter halo guess would have been seen to be questionable. The present paper is solid research that reinforces that view.

This reviewer recommends immediate publication after minor edits in figures.

Comments on the Quality of English Language

Line 132, page 5:  though -> through

Figures 2, 3: labels on the x-axis missing

Author Response

Thank you very much for careful reading and your valuable assessment!

The spelling mistake is corrected, and labels are now added to the x-axis of Figures 2 and 3.

Reviewer 2 Report

Comments and Suggestions for Authors

Report on the paper "The Relevance of Dynamical Friction for the MW-LMC-SMC Triple System"

The paper delves into the dynamics of the Milky Way in conjunction with the Magellanic Clouds, exploring the potential consistency of their trajectories within a 4Gyr timeframe, incorporating the influence of dynamical friction from dark matter.

Regrettably, the current manuscript is plagued by significant issues, primarily pertaining to clarity and understanding of the model and methodologies employed. The following issues must be addressed comprehensively before the paper can be considered for publication in Universe:

Abstract Clarity:

1. The abstract contains a sentence that is both confusing and misleading. The statement regarding the likelihood of the model with friction must be clarified. Is the model likely or unlikely? The significance of this sentence necessitates absolute clarity.

Introduction Coherence. The introduction lacks a structured presentation of past work on the trajectories of the Magellanic Clouds. The authors are urged to:

2. Clearly describe the problem of past trajectories of the MW-LMC-SMC system.

3. Enumerate and order the three to five most crucial approaches to solving the problem.

4. Address the status of the problem in the absence of dynamical friction and justify the interest in dynamical friction as a solution.

5. Conclude the introduction by previewing the analysis's anticipated findings.

6. Provide details on the equation of state used in hydrodynamical simulations, specifying whether dark matter is considered as dust or treated as an ideal gas.

Methodological Clarifications:

7. The choice of rigid NFW profiles over other models needs justification.

8. The last paragraph preceding the conclusions appears to contain key study findings; some of this information should be integrated into the introduction.

9. Clarify the dependence or independence of results on a ±30 percent mass variation.

10. Provide explanations for the parameters used in the MCMC and GA methods.

Minor Comments:

11.Specify acronyms throughout the text and tables.

Once these issues are thoroughly addressed, a revised version of the paper can be reconsidered for further review.

Comments on the Quality of English Language

An overall grammar check would help the paper.

Round 2

Reviewer 1 Report

Comments and Suggestions for Authors

1. The Introduction has been expanded to include other interesting studies done with the dynamical halo friction model.

2. The NFW model has been considerably expanded.

3. The two solution algorithms have been fleshed out with additional references.

4. The actual numerical run results have been expanded more in description.

5. The conclusion is robust and inline with the other researchers who have included dynamical friction.

Reviewer 2 Report

Comments and Suggestions for Authors

The authors have addressed the issues pointed out in my first report. I consider this second version of the paper suitable for publication in Universe.